# Occupational Well-Being of Multidisciplinary PHC Teams: Barriers/Facilitators and Negotiations to Improve Working Conditions

**DOI:** 10.3390/ijerph192315943

**Published:** 2022-11-29

**Authors:** Marta Regina Cezar-Vaz, Daiani Modernel Xavier, Clarice Alves Bonow, Joana Cezar Vaz, Letícia Silveira Cardoso, Cynthia Fontella Sant’Anna, Valdecir Zavarese da Costa, Carlos Henrique Cardona Nery, Helena Maria Almeida Macedo Loureiro

**Affiliations:** 1School of Nursing, Federal University of Rio Grande, Rio Grande 96203-900, Brazil; 2Faculty of Nursing, Federal University of Pelotas, Pelotas 96010-610, Brazil; 3Fundação Getúlio Vargas, Rio de Janeiro 22231-010, Brazil; 4Department of Nursing, Federal University of Pampa, Uruguaiana 97501-970, Brazil; 5Department of Nursing, Federal University of Santa Maria, Santa Maria 97105-900, Brazil; 6Institute of Human and Information Sciences—ICHI, Federal University of Rio Grande—Santa Vitória do Palmar Campus, Santa Vitória do Palmar 96230-000, Brazil; 7School of Health Sciences, Santiago University Campus, University of Aveiro (ESSUA), 3810-193 Aveiro, Portugal

**Keywords:** well-being at work, job commitment, job satisfaction, job involvement, working conditions, risk perception, negotiations, primary health care

## Abstract

Well-being at work is one of the factors determining healthy work conditions and is perceived by workers as a positive psychological state. In this study, the concept of well-being at work was used together with occupational functionality (i.e., current health state, current work environment, and barriers/facilitators to implementing well-being at work), occupational risk perception, and proactivity/negotiations held by workers to improve working conditions. In this context, the objectives were to identify the socio-demographic and occupational characteristics independently associated with levels of well-being at work of the multidisciplinary PHC health team; detect barriers or facilitators resulting from the attitudes of colleagues, community members, and managers that influence the well-being at work of the multidisciplinary health team; and identify with whom and what reasons led health workers to become proactive and negotiate improved working conditions. This cross-sectional study addressed 338 health workers from the multidisciplinary teams of PHC outpatient services in the extreme south of Brazil. Multivariate linear regression models were adopted to analyze data. The results show various independent associations with levels of well-being at work. Nursing workers (technicians and nurses) more frequently expressed job commitment and job satisfaction. Difficulties in solving problems and performing work routines, and co-workers’ attitudes directly influence the well-being of the PHC team members. Risk perception (physical and chemical) also influences well-being. Negotiations in which PHC managers engaged to improve working conditions appeared as a significant predictor of job commitment, job satisfaction, and job involvement. The results reveal that well-being at work is an important indicator of the potential of workers’ proactivity in negotiating improved working conditions.

## 1. Introduction

Occupational well-being is a theoretical model that supports employers and workers to promote occupational health and safety and the global development of the workplace’s structural and managerial aspects to ensure the workers’ health, safety, and well-being [1]. Well-being at work is an important determinant of how workers respond to increasingly complex challenges and demands imposed by the organization and work management, including work requirements, the availability of organizational support, rewards, and interpersonal relationships established in the workplace [1,2].

Well-being at work concerns an individual’s positive and subjective state toward him/herself and organizational behavior. Integrating well-being at work with risk prevention models strengthens occupational safety and health measures within organizations [3,4]. Additionally, well-being at work characterizes the quality of occupational life and is one of the primary determinants of productivity [5,6] and employee creativity [7]. Hence, workers achieve a positive mental state when they experience positive feelings in the workplace (job satisfaction), consonance between their professional skills and demands imposed by tasks (job involvement), and positive feelings toward the organization (affective commitment) [8].

From this perspective, this study’s empirical focus is on the well-being of Primary Health Care (PHC) workers at work. Hence, actions conducive to a healthy work environment within PHC should be highlighted to promote physical, psychosocial, and organizational conditions that favor workers’ health instead of overlapping working processes merely intended to promote profit and production [9]. It is important to note that appreciating the relationship between working conditions, risk exposure perception, and positive well-being at work among multidisciplinary PHC teams contributes to proactively managing occupational health and safety.

The PHC work environment refers to a health care model in force in the Brazilian Unified Health System (SUS). PHC plays a role in coordinating the health care network and integral health care provided to the population [10]. Occupational health, the focus of this study, contributes to organizations determining the health–disease continuum. Occupational health promotes and protects workers’ health and the recovery, rehabilitation, and occupational well-being of workers exposed to risks and injuries in their work environments [1,11]. Therefore, this study focuses on occupational well-being without losing sight of how comprehensive this perspective is, considering the relationship with working conditions. From this perspective, well-being at work can be adopted as a theoretical model in studies addressing the improvement in interpersonal interactions and working conditions. An example would be a study conducted in Spain to analyze the relationship between structural empowerment and psychological empowerment and how they influence well-being at work. Strong and significant influence was reported on improved satisfaction, work environment, and affective commitment to the organizational structure [12].

A study addressing job-related affective well-being among PHC physicians showed that, in most cases, negative emotional perceptions are associated with psychological disorders. For this reason, job-related psychological well-being is considered a motivating factor that minimizes health problems resulting from long working hours, low wages, and unhealthy work environments, which cause stress, anxiety, and depression among PHC physicians. Hence, the more positive and stronger the bonds established in processes and workplaces, the better the workers’ occupational health, the fewer the occupational risks, and the healthier the work interactions [13]. In this sense, well-being at work, especially within PHC [14], is one of the occupational health priorities intended to improve organizational resilience and the working conditions in public health services. This is because a positive work environment that values assessments of affective concepts, such as positive and negative affect, mood, and emotional experiences in the work context, promotes occupational health and the construction of health systems adaptable to organizational resilience [15]. Studies addressing PHC services with a resilient workforce adapted to take on different tasks reported better assessments of leadership, colleagues, salary, opportunities for promotions, and the nature of work, besides higher scores related to commitment and job satisfaction, as well as positive job engagement and job involvement [16,17,18].

Given the tasks resulting from PHC work process related to well-being characteristics, a study conducted in the United Kingdom [2] addressed the management of psychosocial risks at work among occupational health services. The conclusion is that an absence of well-being at work harms individual and collective health and safety because it potentially leads to poor mental health, burnout, heart diseases, and musculoskeletal disorders, besides other adverse organizational outcomes (e.g., absenteeism, decreased productivity, and human error). The International Labor Organization [19] and the World Health Organization [20] consider these disorders as socio-environmental phenomena produced by working conditions. Hence, organizations and workers must negotiate agreements to modify local, regional, and global policies prioritizing employee well-being as an indispensable tool in promoting healthy work environments. According to the literature [21,22], a healthy work environment defines the current labor market because there is an attempt to promote workers’ well-being by taking into account demographic changes, economic slowdowns, and impacts on the population’s health caused by events such as the COVID-19 pandemic, for instance, and the aging of the working population, coupled with rapid technological change.

Few studies address well-being at work in the health field, especially within PHC, considering a theoretical model integrating the three most important concepts (i.e., commitment, job satisfaction, and job involvement). Additionally, other essential concepts can strengthen well-being by integrating working conditions represented by risk perception and occupational functionality, the current state of health and current work environment, and the barriers and facilitators resulting from the attitudes of colleagues and community members and negotiations to improve working conditions. This set of elements comprises the most comprehensive characteristics of occupational well-being while considering its socially complex features. The main concepts are presented below.

Using a combination of different concepts surrounding well-being, this study’s objectives include: identifying the socio-demographic and occupational characteristics independently associated with the indices of well-being at work of a multidisciplinary PHC health team; detecting barriers or facilitators resulting from the attitudes of colleagues, community members, and managers that influence the well-being at work of a multidisciplinary health team; and identifying with whom and for what reasons health workers become proactive and negotiate improved working conditions.

## 2. Theoretical Framework

### 2.1. Well-Being at Work

According to Ryan and Deci [23], well-being concepts are grouped into two perspectives: one addressing a subjective state of happiness, which corresponds to subjective well-being, and one focusing on human potential, which is equivalent to psychological well-being. These two conceptual perspectives are intertwined with two philosophical accounts of happiness, hedonism and eudaimonism. In the first account, well-being means pleasure and happiness (hedonic well-being). In the second account, well-being consists of an individual reaching his/her full potential, ability to think, reasoning, and common sense (eudaimonic well-being) [23].

Howell et al. [24] note that “well-being” refers to a positive mental health state. From this perspective, the concept of well-being at work is a positive mental state arising from workers experiencing positive feelings due to the work environment’s characteristics (job satisfaction) [25], finding consonance between one’s professional skills and the demands imposed by tasks (job involvement) [26], and positive feelings towards the organization (affective organizational commitment) [27]. The theoretical model of these three integrated concepts comprises psychological aspects of a cognitive (mental) nature, in which strictly positive beliefs and feelings that emerge from the organizational work context are considered. Job satisfaction is one’s level of contentment with work relationships, leadership, colleagues, tasks, and reward systems involving salaries and promotions. Job involvement concerns one’s identification with work, and affective organizational commitment refers to workers’ positive and negative feelings towards their organizations [25,26,27].

Therefore, the concept of occupational well-being integrating these three concepts is adopted in this study. It is structured using two factors: Commitment and Job Satisfaction, the latter of which contains in its propositional core positive bonds established with the organization (affective commitment) and satisfaction with the management, salary, and promotion opportunities, for example [25,27]. In turn, the Job Involvement factor refers to a belief that one’s work elicits pleasant hours at work and contains essential aspects of life [26]. This conceptual conjunction of occupational well-being establishes this study’s outcome.

### 2.2. Occupational Functionality

Occupational well-being has the potential to affect occupational functionality. Functionality conditions the workforce to feel healthy or unhealthy and directly influences one’s ability to perform tasks. According to the International Classification of Functioning, Disability, and Health (ICF), functionality is understood as an attribute that involves bodily functions, tasks, and participation and identifies the workplace as a facilitator or a barrier promoting or hindering the performance of tasks and the participation of people in this environment [28].

From this perspective, the ICF can guide the performance of health workers in the workplace as it classifies health-related states, ensuring that workers’ physical and psychological dimensions are monitored. It also can help analyze how the workplace influences workers’ health, providing a socio-environmental view of the relationship between the workplace and the professional health state.

This paper focuses only on occupational functionality, considering the concepts of the current workplace and health state to perform tasks from the workers’ perspective. Likewise, functionality allows us to identify the attitudes of colleagues and the community as potential barriers or facilitators.

The health state refers to how an individual perceives her/his biopsychosocial health condition in a given environment. In this study, the current environment corresponds to the work environment, which is physical, social, and attitudinal. It is important to note that ICF considers the environment to be the context in which a person describes a given situation. In other words, how an individual perceives his/her physical and psychosocial conditions to perform routine tasks in the workplace. It is important to remember that the ICF adopts a more comprehensive classification than the one used here. However, the concepts provided theoretical support to interpret the occupational well-being phenomenon. These concepts are presented in the form of variables in the Methods section.

### 2.3. Occupational Risk Perception and Negotiations to Improve Working Conditions

According to the theoretical orientation of Slovic (2000) [29]; Sjöberg (2000) [30]; and Sjöberg, Moen, and Random (2004) [31], this study assumed that the notion of risk perception involves two aspects: the magnitude of a potential loss and the probability of its occurrence [32]. In other words, the existence or inexistence of different risk factors explains why people perceive the same risk in very different situations or why the same individual may perceive a risk differently depending on when s/he is asked about it [33]. Risk perception is a multifaceted phenomenon that includes social, cultural, and political aspects in its production and reproduction [32,34,35]. Based on this theoretical orientation, the researchers observed the coherence and need to analyze occupational well-being, relating it to the multidisciplinary PHC health team’s risk perception.

Considering that risk perception encompasses personal ideas and constructs as well as those related to the workplace, to perceive risk, one needs to believe in it [30]. Therefore, identifying the risk perceptions held by PHC workers is relevant because these individuals are co-responsible for the risks perceived in the workplace and can attempt to prevent perceived risks for their sake and that of the collective workforce. Therefore, there is the possibility to change, minimize, or even eliminate risk factors related to individual and collective behaviors, promoting better working conditions.

Knowledge about working conditions is directly related to risk perception. Hence, this study’s concept of perception represents the workers’ knowledge about their work environment and its conditions. Thus, one of the interaction processes to promote changes is negotiating for improved working conditions to promote well-being in the workplace. Negotiation is pro-active; that is, it is a critical action that anticipates preventive measures to solve problems perceived as an individual and/or collective need in the workplace. Proactive negotiation contains different sources of support and decision-making, such as colleagues, managers/supervisors, labor unions, community members, and other potential partnerships to join efforts to improve working conditions. From this perspective, negotiation involves workforce proactivity in participating in the improvement in PHC working conditions.

Furthermore, the interest in integrating the concepts of risk perception and negotiation into the outcome of interest is due to a belief that risk perception reflects working conditions and must be explicitly stated in negotiations promoted by health workers to improve working conditions. Risk perception refers to technological knowledge to provide occupational health education. The ability to disseminate knowledge and apply technologies to prevent risks in public health can influence occupational well-being among PHC workers. Workers should be encouraged to apply their knowledge regarding their well-being and a healthy future work environment. Additionally, having knowledge and scientific knowledge regarding individual and collective behavior contributes to negotiations and decision-making regarding working conditions, promoting workers’ well-being and well-being at work. Moreover, changing or promoting awareness about health, illness, and work can be improved in the learning process to direct perceptions of what may or may not influence or even determine injuries, illnesses, or better health conditions among workers and in the work environment. Negotiation enables workers to make informed decisions and adopt behaviors that promote their health and well-being.

## 3. Materials and Methods

### 3.1. Study Design and Participants

It is important to note that this study on well-being at work is part of a larger project entitled Socio-environmental Dimension of the Health of PHC Workers in Southern Brazil (“Dimensão Socioambiental da Saúde do Trabalhador da Atenção Básica de Saúde no Sul do Brasil”) conducted by a team of researchers from the Laboratory of Socio-Environmental Processes Studies and Collective Production of Health (LAMSA) affiliated with different universities in southern Brazil. This cross-sectional study complied with the Declaration of Helsinki, and the Institutional Review Board (Conep) at the Federal University of Rio Grande approved its protocol (CAAE: 70043717.0.0000.5324). The objective was to investigate the relationship between the well-being of PHC workers and negotiations to improve working conditions by integrating occupational functionality and considering barriers, facilitators, and occupational risk perception. It was conducted in two cities in the extreme south of Rio Grande do Sul, Brazil. City 1 has medium-sized characteristics and has 31 outpatient PHC units, while City 2 is small and has 10 outpatient PHC outpatient units.

The sample size was calculated using the Epi Info^®^ StatCalc tool (version 7.2, CDC, Atlanta, GA, USA). We considered 548 professionals working in PHC during the study period, including nurses, doctors, dentists, nursing technicians, community health agents, and oral health assistants/technicians. We considered a margin of error of 5%, 95% confidence level, and 5% of losses. The professionals were selected through non-probabilistic sampling. The consecutive intentional sample [36] was supposed to comprise at least 232 professionals from the coverage area. The inclusion criterion was working in PHC for at least six months, and the exclusion criterion was being on leave during data collection from January to March 2020. A total of 342 health professionals were interviewed [37]; however, because the well-being at work scale was incomplete in 4 interviews, 338 interviews were included in the analysis. The participants were distributed as follows: 50 nurses, 43 doctors, 72 nursing technicians, 139 community health agents, 13 dentists, 15 technicians/assistants of oral health, and 6 others. “Others” corresponds to PHC workers included in some health teams (1 physical therapist, 1 physical educator, 2 psychologists, and 2 nutritionists). These workers were included because of the characteristics of these health teams. Considering the representativeness of each profession and the complexity and diversity of actions developed by primary outpatient care workers, we chose to keep adding participants to the sample until the last day of the data collection. This decision was first based on work similarities and differences, which were carefully addressed in the data analysis. The responsibilities of the professionals in the PHC teams followed the legal provisions regulating each profession [38]. Community health agents (CHA) are professionals mediating technical and common knowledge and linking the health team to the community, facilitating and promoting the population’s access to health services [10]

Healthcare professionals were recruited and interviewed from January to March 2020. Face-to-face interviews were held at their workplaces by previously trained researchers. Two or three researchers always worked together to ensure their safety and speed up the selection process. At the time, the university’s teaching and research activities had not yet been suspended due to the COVID-19 pandemic. Hence, the individual interviews with the PHC workers were face-to-face and lasted 58 min on average. All participants received clarification about the study’s objectives and signed two copies of informed consent forms before participating.

### 3.2. Measures

A structured form addressed socio-demographic information (i.e., age, self-reported race, marital status, education, and the number of children), PHC job (i.e., place of work, whether the participant had a second job, type of PHC unit, profession, years of professional experience, years working in a PHC service, weekly working hours, PHC unit’s working hours, and monthly income). The following variables were used to address the specificity of occupational functionality according to the ICF: health state and current work environment concerning difficulty in solving problems, performing routines, and multiple work tasks; barrier and facilitator levels resulting from the attitudes of colleagues and community members. We included in the negotiation for improved working conditions those with whom negotiation was conducted (source of negotiation: colleagues, managers/supervisors, community members, or labor unions) and the reasons to negotiate. A dichotomous variable was adopted to define occupational risks (i.e., physical, chemical, ergonomic, biological, and psychological) and measure risk perception.

The Inventory Well-Being at Work—(IWBW-13) (Inventário de Bem-estar no Trabalho (IBET-13)) [39,40,41] was applied to assess the levels of positive psychological states in the work environment (occupational well-being/well-being at work). Detailed overview of the Inventory Well-being at Work can be found in the Appendix A. The variables of the structured form (i.e., socio-demographic and occupational variables, occupational functionality, negotiation, and risk perception) were tested and adjusted during meetings held with LAMSA members and through a pilot study applied to ten individuals from different professions in the health field before data collection. The questions selected from the individual questionnaire applied to Primary Health Care workers can be viewed in the Appendix A. This pilot study’s primary objectives were to assess and adapt the data collection instrument regarding its effectiveness and comprehension—that is, how easy or difficult the questions were and how to improve the field researchers’ interviewing skills [37]. Figure 1 presents the main concepts integrated into well-being at work (occupational functionality, risk perception, and negotiation to improve working conditions) and respective qualifiers to investigate the magnitude of the problem among professionals in the multidisciplinary PHC health teams.

The Inventory Well-Being at Work—(IWBW-13) (Inventário de Bem-estar no Trabalho (IBET-13)) [39,40,41] contains 13 items that include components of the constitutive psychological model and classic concepts of organizational behavior, which are important elements to define well-being at work: affective organizational commitment, job satisfaction, and job involvement. It was designed to capture the workers’ positive state of well-being at work. The Inventory Well-being at Work was validated in research addressing its conceptual structure, with potential application in Brazilian studies [40,41], among other investigations. These studies confirm that the subject-centered Inventory Well-being at Work is consistent, sensitive, and well-accepted by respondents. According to its conceptual structure and applicability, it contains two factors/dimensions. Dimension 1 comprises the Commitment and Job Satisfaction factor, expressing positive feelings toward the organization—that is, affective commitment and satisfaction with management, salary, promotions, and tasks. Dimension 2 comprises the Job Involvement factor, which concerns beliefs that the work performed provides pleasant hours and contains important aspects of life. Nine items describe the Commitment and Satisfaction dimension, while the Job Involvement dimension is described by four items. To obtain the Inventory Well-being at Work scores, we need to sum the scores obtained in each item and divide them by each factor’s number of items. The healthcare professionals were asked to rate the 13 items on a 5-point scale (1 = Strongly Disagree, 2 = Disagree, 3 = Neither Agree nor Disagree, 4 = Agree, and 5 = Strongly Agree). Thus, the scores assigned to the nine items of Dimension 1 (Commitment and Satisfaction) must be summed and divided by 9. As for Dimension 2 (Job Involvement), the scores assigned to its four items are summed up and then divided by 4. In the end, two average scores are obtained. These scores are interpreted considering that scores between 4 and 5 are considered high scores; between 3 and 3.9 are considered moderate scores, and scores between 1 and 2.9 are considered low scores. The same interpretation was applied to the well-being at work general score. It is important to note that the items comprising a given factor were considered when interpreting the psychological content represented by the average score. The use of the Inventory Well-being at Work to measure well-being at work is explained by its psychometric potential and application, which decreases the number of items to measure the positive mental state of well-being at work together with other concepts, as was the procedure here (Appendix A).

Given a consensus in the literature addressing psychometric properties that Cronbach’s alpha is not recommended if there is uncertainty regarding an instrument’s dimension or suspected multi-dimensions, experts suggest that McDonald’s omega be used to assess an instrument’s reliability. This coefficient is recommended due to its good performance in breaking assumptions for Cronbach’s alpha [42]. In this study, the McDonald’s omega obtained by the Commitment and Satisfaction factor was 0.82, that of the Job Involvement factor was 0.65, and the total scale 0.86. These reliability measures are usually above 0.7; thus, only the Job Involvement factor obtained a lower score. However, according to Altman (1991) [43], a reliability coefficient above 0.6 would already be satisfactory, considering the Job Involvement factor presents a lower number of items (4). Thus, the scale presented good internal consistency.

In this study, occupational functionality is a variable that corresponds only to the current work environment and health state to perform tasks (solving problems, performing multiple tasks, and the work routine) based on the workers’ self-perception. Likewise, functionality enables identifying the attitudes of colleagues and community members as potential barriers or facilitators for hindering or promoting well-being at work. As previously mentioned, the individual perceives his/her physical and psychosocial conditions to perform daily tasks in the work environment (Appendix A).

The Occupational Health and Safety Act [44], which classifies occupational hazards as physical, chemical, biological, physiological, or psychosocial, was used to analyze risk perception [29]. Physical hazards include noise, vibration, ionizing and non-ionizing radiation (ultraviolet and infrared radiation), and electromagnetic fields. Chemical hazards include exposure to lead, mercury, benzene, asbestos, and materials containing such chemicals. Biological hazards are microorganisms such as bacteria, viruses, and fungi. Ergonomic hazards include intense physical exertion, repetitive movements, and physical positions and work postures that cause fatigue. Finally, psychological risks refer to monotonous work or work not suited to a worker’s abilities, poor work organization, and working alone for long periods (Appendix A).

Finally, negotiation to improve working conditions considered sources, such as colleagues, community members, managers/supervisors, occupational safety workers, and the labor union. The reasons for negotiating were also considered and included encouraging employee compliance with Personal Protective Equipment (PPE); modifying exposure to occupational risks (i.e., physical, chemical, ergonomic, biological, and psychological); promoting discussions/changing work goals; conquering the right to health at work; discussing working hours; improving interaction/communication among workers; salary; improving interaction/communication between workers and employers, and discussing resources provided to unions (Appendix A).

### 3.3. Data Analysis

Statistical analyses were performed using SPSS v. 21.0 (IBM Corp., Armonk, NY, USA). The quantitative variables were described using means and standard deviations or medians and interquartile intervals. The categorical variables were described using absolute and relative frequencies. Student’s *t*-test for independent samples or analysis of variance (ANOVA) complemented by Turkey’s test was applied to compare the means. Pearson’s correlation coefficient or Spearman’s correlation tests assessed associations between continuous and ordinal variables.

A Multivariate Linear Regression analysis with the Backward model extraction method was used to control for confounding factors. First, the regression or angular coefficient (b), which measures the effect of increasing one unit in the factor on the outcome, was calculated with a 95% confidence interval (Appendix A). Additionally, the standardized beta coefficient (β) was also presented to compare the strength of the association between the variables in the multivariate model, as it does not have a unit of measurement; the higher the coefficient, the stronger the association. Finally, the coefficient of determination (R2) was calculated to determine what percentage of the variation in a specific outcome is explained by the multivariate model. The criterion used to include a variable in the model was a *p* < 0.20 in the bivariate analysis, and the criterion for it to remain in the model was a *p* < 0.10 in the final model (criterion used in the Backward method); the significance level was set at 5% (*p* ≤ 0.05).

## 4. Results

### 4.1. Descriptive and Bivariate Analysis

#### 4.1.1. Socio-Demographic and Job-Related Variables

The participants comprised 338 PHC workers at the outpatient level providing care to the communities in two cities. City 1 has medium-sized characteristics with 31 PHC outpatient units, while City 2 is small and has 10 PHC outpatient units. The participants were 41.4 (±9.9) on average; most (86.6%) were women; married/consensual union (58.6%); and self-reported Caucasian. Community health agents (41.1%), nursing technicians (21.3%), and nurses (14.8%) comprised the largest group of professionals in the sample, while most (81.6%) did not have a job other than at the PHC service (Table 1).

As previously noted, the Inventory Well-Being at Work is composed of two dimensions: Commitment and Job Satisfaction and Job Involvement. It is noteworthy that the mean obtained in the Commitment and Job Satisfaction dimension was 2.83 (±0.72), while 2.65 (±0.72) points were obtained in the Job Involvement dimension; the general score was 2.74 (±0.64). Figure 2 shows that low scores were more frequently obtained in the Job Involvement dimension. It is important to note, however, that low scores were obtained in both dimensions. According to the IWBW-13 rating, 4 and 5 indicate a high score, between 3 and 3.9 indicates a moderate score, and between 1 and 2.9 indicates a low score.

Table 1 presents the associations between socio-demographic and job-related variables with the well-being at work (Commitment and Satisfaction, Job Involvement) dimensions. The bivariate analysis of those dimensions showed associations with different variables that were considered to be potentially influenced by the PHC professionals’ positive states, i.e., their well-being at work. The highest scores obtained in Commitment and Job Satisfaction were significantly associated with widowed professionals (*p* = 0.038). As for the Job Involvement factor, the highest scores were significantly associated with being a man (*p* = 0.016), having a second job (*p* = 0.010), being a technician or nursing assistant (*p* = 0.005), and having fewer years of experience in PHC (*p* = 0.033).

#### 4.1.2. Well-Being at Work and Occupational Risk Perception

PHC workers who perceived physical and chemical risks in their workplace obtained lower well-being scores in both dimensions, i.e., Commitment and Job Satisfaction ((*p* = 0.012, *p* = 0.006, respectively) and Job Involvement (*p* = 0.020, *p* = 0.003, respectively). Additionally, the perception of ergonomic risks (*p* = 0.040) was associated with lower scores only in the dimension of Commitment and Job Satisfaction; workers who perceived psychosocial risks (*p* = 0.001) and obtained the lowest well-being scores at work were associated with the dimension of Job Involvement (Table 1).

The interpretation of these findings reveals that biological risks, frequently perceived in PHC work environments, were not statistically associated with well-being at work. Such a fact becomes evident considering the number of health professionals who perceived biological risks (*n* = 317) and those who perceived themselves as not being exposed to biological risks (*n* = 20). The relationship between health/safety and well-being at work demands attention, so it is not irrelevant to well-being in the work environment; i.e., the perception of occupational risk is an important indicator of well-being at work. In this context, the perception of physical, chemical, ergonomic, or psychosocial risks more clearly indicates well-being than the perception of biological hazards. Thus, this analysis is relevant to confirm that occupational risks are present and perceived as a condition for well-being at work from PHC workers’ perspective.

#### 4.1.3. Well-Being at Work and Occupational Functionality

Proceeding with the bivariate analysis, Table 2 shows that the highest scores obtained in the Commitment and Job Satisfaction and the Job Involvement dimensions were significantly associated with workers’ functionality (ICF variables). In other words, workers with higher well-being scores reported lower difficulty in solving problems and performing tasks and work routines while considering their health state and current work environment. It is important to note that these results refer to actions that are necessary for the current work environment (e.g., solving problems, performing multiple tasks, and performing work routines) while considering the workers’ current health state (health perception), which represents occupational functionality; the two dimensions of well-being were associated in this initial analysis.

In the representation of the ICF variables, Table 2 shows that the scores assigned by health professionals to the attitudes of community members and colleagues were associated with the highest scores of Commitment and Job Satisfaction for those who identified lower barriers for both attitudes (*p* < 0.001). As for the scores assigned to the attitudes of community members and colleagues, higher scores of Job Involvement were obtained by professionals who perceived lower barriers in the community members’ attitudes (*p* = 0.044); however, no association was found for colleagues’ attitudes. Confirming these associations requires paying attention to the process because workers might improve and deepen their understanding of the aspects of a positive psychological state at work—occupational well-being.

On the other hand, the bivariate analysis revealed no association between the attitudes of community members or colleagues as facilitators of Commitment and Satisfaction and Job Involvement. However, many respondents indicated that the attitudes of community members and colleagues worked as facilitators at some level. These results indicate that the attitudes of community members and colleagues promote the flow of work, and any interference might have a negative potential and decrease health workers’ well-being at work. Hence, facilitating such a flow seems beneficial, while the opposite prevents this “flow” from contributing to the workers’ positive state.

#### 4.1.4. Well-Being at Work and Negotiations to Improve Working Conditions

Regarding negotiations to improve working conditions of the PHC environment, as a factor determining well-being, the highest Commitment and Satisfaction scores were obtained by workers who negotiate with managers/supervisors (*p* < 0.001), community members (*p* = 0.001), and occupational safety workers (*p* = 0.005). The highest scores in the Job Involvement factor were obtained by workers who do not negotiate for workers (*p* = 0.038) but negotiate for managers/supervisors (*p* < 0.001) and occupational safety workers (*p* = 0.027). Among the potential factors influencing the well-being of health workers, not negotiating (exchanging and establishing agreements or bargains) with colleagues increases occupational well-being (Table 3). However, this finding was not confirmed in the multivariate analysis, as shown in the following section.

The reasons for why negotiating improved working conditions, from the participants’ perspective, are presented below. Table 3 and Figure A1 (Appendix B) present the sample distribution according to the reasons for negotiating improved working conditions (health and safety). The reasons most frequently mentioned were: salary (59.8%), discussing/changing work goals (58.0%), and resolving conflicts of interest (51.2%).

Regarding the reasons for negotiating improved working conditions (health and safety) to promote occupational well-being, changing exposure to ergonomic occupational risks (*p* = 0.041), occupational accidents (*p* = 0.010), psychosocial risks (*p* = 0.001), arranging for the acquisition of PPE (*p* = 0.049), improving salaries (*p* = 0.001), resolving conflicts of interest (*p* < 0.001), redistributing working shifts (*p* = 0.042), and changing working hours (*p* = 0.022) were significantly associated with lower Commitment and Satisfaction scores. Changing exposure to occupational accidents (*p* = 0.045), changing exposure to psychosocial risks (*p* = 0.040), and redistributing working time (*p* = 0.034) were significantly associated with lower Job Involvement scores.

### 4.2. Multivariate Linear Regression Analysis

#### 4.2.1. Well-Being at Work According to the Commitment and Satisfaction Dimension and Independent Factors

After making adjustments according to the multivariate model for potential confounding factors (Table 4), the following independent factors remained statistically associated with high scores: health workers from City 1 (*p* = 0.005), being a nurse (*p* = 0.003), being a nursing technician/assistant (*p* = 0.002) or a dentist (*p* = 0.015), having less experience at a PHC (*p* = 0.027), a perception of not being exposed to physical (*p* = 0.010) or chemical (*p* = 0.021) occupational hazards, lower level of difficulty in solving problems in the current environment (*p* < 0.001), assigning a lower score to the colleagues’ attitudes regarding barriers (*p* = 0.017), and negotiating improvements with managers/supervisors (*p* < 0.001) and community members (*p* = 0.001). On the other hand, being a physician (*p* = 0.040) and the reason for negotiations being the redistribution of working shifts were significantly associated with lower scores. This model explains 36.4% of the variability inherent to Commitment and Satisfaction scores.

A deeper analysis shows that health professionals in City 1 present an average increase of 0.26 points in the Commitment and Satisfaction scores (95% CI: 0.08 to 0.44). On average, each additional year working in a PHC service decreases the score in this outcome by 0.01 points (95% CI: −0.02 to −0.00). Nurses, nursing technicians/assistants, and dentists scored 0.31 (95% CI: 0.11 to 0.51), 0.27 (95% CI: 0.10 to 0.45), and 0.42 (95% CI: 0.08 to 0.76) points higher than community health agents in this factor, respectively. However, medical professionals scored 0.23 (95% CI: −0.44 to −0.01) points below the community health workers’ scores in this dimension.

On average, the workers who perceive physical and chemical occupational hazards present a decrease of 0.32 (95% CI: −0.56 to −0.08) and 0.21 (95% CI: −0.39 to −0.03) points in this dimension, respectively. Furthermore, each higher level of perceived difficulty in solving a problem in the current environment decreased the Commitment of Satisfaction scores by 0.15 points on average (95% CI: −0.23 to −0.07). Additionally, each additional level in the scores assigned to the colleagues’ attitudes concerning barriers decreased the Commitment and Satisfaction scores by 0.09 points on average (95% CI: −0.16 to −0.02). Finally, the workers who negotiate improvements with managers/coordinators and community members presented an increase of 0.44 (95% CI: 0.31 to 0.58) and 0.40 (95% CI: 0.17 to 0.63) points on average in this factor, respectively.

Finally, the reason for negotiation being the redistribution of working time remained significantly associated with lower scores in this factor; that is, this reason for negotiation led to an average decrease of 0.23 points in this factor’s scores (95% CI: −0.44 to −0.02). Thus, workers negotiate this matter when they feel the need; that is, they dialogue when experiencing a decrease in their positive state at work (Table 4).

Hence, the standardized regression coefficient (beta) confirms that the variables most strongly associated with Commitment and Satisfaction scores were negotiating for improvements with managers/coordinators and difficulty in solving problems in the current work environment.

#### 4.2.2. Well-Being at Work, the Job Involvement Dimension, and Independent Factors

Regarding the Job Involvement dimension (Table 5), after adjustments, the following remained statistically associated with high levels of Job Involvement: being a nursing technician/assistant (*p* < 0.001), night-time shift (*p* = 0.042), night-time/daytime shift (*p* = 0.028), not perceiving exposure to chemical occupational risks (*p* = 0.001), lower level of difficulty in performing work routines in the current work environment (*p* < 0.001), and negotiating for improvements with managers/coordinators (*p* < 0.001). This model explains 24.4% of the variability inherent to Job Involvement scores.

These results confirm that in this factor, nursing technicians/assistants scored 0.46 (95% CI: 0.28 to 0.64) points higher than community health agents. In addition, health professionals working night-time/daytime scored 0.64 points higher in Job Involvement than those working other shifts (95% CI: 0.00 to 1.28). Health workers who perceived exposure to chemical risks present a decrease of 0.32 (95% CI: −0.51 to −0.14) points on average in this factor. Each additional level of perceived difficulty in performing work routines in the current environment decreased the Job Involvement (95% CI: −0.25 to −0.09) scores by 0.17 points on average. Professionals who negotiated improvements with managers/supervisors showed an average increase of 0.39 (95% CI: 0.25 to 0.54) points in this factor. Furthermore, the reason for negotiations other than addressing the redistribution of working time (*p* = 0.030) remained associated with high scores in this dimension. However, in the following inverse interpretation, when the reason for negotiations is to address the redistribution of working shift, this factor’s scores decrease by 0.25 points on average (95% CI: −0.48 to −0.03) (Table 5).

The standardized regression coefficient shows that the variables most strongly associated with the Job Involvement dimension were negotiating improvements with managers/supervisors and being a nursing technician/assistant.

### 4.3. Synthesis of the Occupational Well-Being of the Multidisciplinary PHC Team

This analysis indicates distinctive elements related to the concepts adopted here. First, nursing technicians/assistants, nurses, and dentists more frequently reported well-being at work in terms of the Commitment and Satisfaction and Job Involvement dimensions. However, after adjusting for confounding factors, only nursing technicians/assistants remained associated with the highest scores in both dimensions. The nurses remained associated with the Commitment and Satisfaction dimension. Thus, the PHC nursing staff presented the highest well-being at work. Regarding work-related characteristics, the length of time working at the PHC and night-time and night-time/daytime shifts were elements of the work organization that tended to lead workers to obtain the lowest well-being scores at work.

The perception of exposure to occupational risks (physical and chemical) stood out as a factor that decreases well-being scores in the work environment. According to the analysis design, this element (risk perception) is part of the PHC working conditions, leading us to highlight the strong influence of the conditions in which PHC practices are operationalized on the well-being of its workforce. Regarding occupational functionality, when the workers’ ability to perform work routines and solve work problems decreases, their well-being at work also decreases. The co-workers’ attitudes remained associated with barriers to the Commitment and Satisfaction dimension, decreasing well-being at work.

Negotiating for improved working conditions remained strongly associated with managers/supervisors for both components of well-being at work. Association with community representatives was found only for the Commitment and Satisfaction dimension. The results indicate that workers’ ability to negotiate with managers/supervisors increases well-being. The main reason negatively associated with well-being at work was the redistribution of working hours as a representation of the organization of PHC work. Finally, negotiation represents workers’ proactive behavior to establish partnerships to improve working conditions and obtain well-being at work in PHC services.

## 5. Discussion

In this study, the bivariate analysis indicated that the Commitment and Job Satisfaction factors positively influenced affective commitment to the organization among married individuals. However, this variable did not remain independent of well-being at work after adjustments. The same was found in studies addressing well-being among community health workers in China [45] and the United Kingdom [46]. The studies showed positive commitment and job satisfaction among married professionals, which aligns with the result reported here regarding positive commitment and job satisfaction. This finding is possibly explained by the cultural contexts of these two countries, China and Brazil, regarding values and beliefs concerning the relevance of work in people’s lives and its importance for sustaining the family, for instance.

As for the multidisciplinary team, this study shows that the nursing team obtained the highest scores in the perception of well-being at work; being a nurse or a nursing technician remained positively associated with commitment and job satisfaction. In addition to the nursing team, dentists also expressed well-being at work in this factor. On the other hand, physicians obtained lower scores in the Commitment and Job Satisfaction factor. It is important to note that nursing technicians also expressed well-being at work through job involvement. Hence, these results suggest that nursing technicians more strongly perceive well-being in the work environment. Some studies addressing well-being at work in the health field show that nurses [47], physicians [48,49], and community health agents [50] present higher levels of commitment and job satisfaction and positive affective bonds with their organizations when they feel appreciated through the receipt of bonuses, prizes, remuneration, and benefits, in addition to enjoying better working conditions. This may occur because PHC nurses usually carry out work processes encouraged by exchange rewards; that is, they are encouraged by gratifications, which have the potential to channel nurses’ commitment to achieving the goals of the PHC health organization and experiencing satisfaction and positive affective bonds that inspire them to maintain the quality of the service provided to the population.

As opposed to this study, the results reported by a study conducted in Brazil [43] show that nursing technicians do not always experience commitment and job satisfaction or establish affective bonds with health organizations. The reason for this is that they are not always satisfied with their supervisors, salaries, or the few bonuses received. However, the findings reported by the study mentioned earlier concerning job involvement corroborate this study’s findings to the extent that job involvement stems from nursing technicians’ perceptions of their importance to the health care provided to the population. Such a perception leads them to experience pleasant hours while performing their tasks, which positively impacts their well-being at work and in life.

As for the working conditions represented by the health professionals’ risk perceptions, greater commitment and job satisfaction and positive bonds with the organization are reported when workers perceive that they are not exposed to physical or chemical hazards. These workers also report greater job involvement when they perceive that they are not exposed to chemical hazards. In this sense, risk perception should be encouraged to be used together with the concept of well-being. This integration refers to different nuclei of work environment management, one for managing risks in the work environment and the other for producing occupational well-being. Both complement each other because they direct the approach to promote healthy environments conducive to occupational health and safety, focusing on well-being at work.

Not perceiving exposure to biological hazards did not appear as an indicator of well-being at work, which is an expected finding considering that such hazards are more frequently acknowledged and considered to belong to the work process in the health field. On the other hand, the results concerning a perception of not being exposed to physical or chemical hazards is an important way to assess well-being at work. Even though biological hazards represent an occupational risk, they may be trivialized at this level of primary care because they are expected in this environment and are not unfamiliar. However, their “unreal” absence does not indicate well-being at work. The literature shows that commitment and job satisfaction increase the positive bonds of health professionals with the work environment to the extent that job involvement increases when professionals learn to decrease occupational risks, which contributes to greater job involvement and pleasant working hours [51,52,53,54,55].

In the same sense, a study conducted in India [56] and Egypt [57] with health professionals working in surgical rooms showed that perceived occupational risks are usually considered deleterious to job satisfaction in the health field. The complexity of the work environment in the health field is determined by various occupational hazards ranging from accidental, physical, chemical, biological, ergonomic, and psychosocial risks. However, the surgical environment’s physical, chemical, and psychosocial risks were those most frequently perceived, making health workers more vulnerable. Therefore, negative ties are established with the health organization, causing dissatisfaction and a lack of commitment toward work tasks.

Additionally, health workers experience positive commitment and job satisfaction towards the organization and job involvement when they perceive lower difficulty in solving problems and performing multiple tasks and work routines, considering their health state and the current work environment (functionality). This means that the lower the difficulty in solving problems, the greater one’s commitment and job satisfaction, indicating that the organization needs to support workers to solve problems and keep them committed and satisfied. This leads to a broader concept, rather than merely performing the activities in the work process, because the workforce needs to be empowered. Some studies addressing nursing workers in hospitals in Saudi Arabia [58], Egypt [59], and Greece [60] showed that empowering workers favors positive bonds with the health organization, affective commitment, job satisfaction, and job involvement.

These studies refer to empowerment as a strong predictor of job involvement, contributing to the advancement, learning, and innovation of health work processes. These studies also reinforce the evidence that health professionals become more involved with their work as they perceive it to be positive and pleasant and feel that they participate in management processes (e.g., delegating decisions, promoting training programs, and investigating opportunities for improving self-care and self-confidence). It is also noteworthy that professional experience facilitates access to the information necessary to implement effective decisions that contribute to solving problems, coping with working hours and various functional activities, and experiencing well-being at work as a result. An opposite finding was shown in this study regarding less-experienced workers. However, attention should be paid to this finding because it is possibly related to what was said earlier (on average, each additional year working in a PHC service decreases the well-being at work score by 0.01 points), though it was not directly investigated here. It is possible that more experienced workers need opportunities to improve self-care and self-confidence, considering they already have experience and knowledge.

It is also worth noting that the bivariate analysis showed that the lower the barriers concerning the attitudes of community members and colleagues, the higher the health professionals’ commitment and job satisfaction. The only relationship found for job involvement was that lower barriers concerning the community members’ attitudes led to higher levels of well-being. It is important to note that the community members’ attitudes can influence levels of well-being at work, though the multivariate linear regression did not confirm this. Levels of barriers appeared to be associated with colleagues’ attitudes only in the Commitment and Job Satisfaction factor. The literature shows the importance of developing and establishing healthy relationships and environments with colleagues at work [1,9,61,62,63,64]. However, the relationships within the workforce may elicit organic changes, which inhibit well-being in the work environment, such as depression, anxiety, and burnout, which may compromise the functionality of health professionals [28,65,66,67]. A study conducted in health centers in Asia [68] with health professionals responsible for preventing and controlling infectious diseases presented relatively low rates of job satisfaction and affective commitment toward the health institution. However, the authors report the contributory attitudes of co-workers as a facilitator of the work process and improvement in work well-being, considering that interpersonal relationships and respect at work were associated with increased motivation for collective work.

Low well-being scores were found in this study. Additionally, the vast majority of PHC health team professionals considered the attitudes of community members and colleagues to be facilitators. However, it was not possible to establish an association between facilitator degrees (attitudes of community members and colleagues) with rates of well-being at work. Nevertheless, we would like to reinforce that the job involvement of health workers motivated to work with the community positively influenced qualified working hours in the work process. In this context, the health professionals’ job satisfaction and involvement positively influenced the actions concerning health promotion, the prevention of diseases and epidemics, and treatments and rehabilitation of individual and collective health [69,70].

Furthermore, integrating health workers and community members enhances healthy conditions in the work environment. An example of such interaction is presented by a study addressing health professionals providing primary care in China [45]. Similar to this study, it shows other results besides job commitment related to community support. The Chinese study presents family support as another relevant factor affecting job involvement among health professionals providing primary care. It also indicates the need for management services to support primary care providers by implementing labor intervention processes that include support not only from the community but also from families, organizations, society, and government. The greater the health professional’s sense of social, organizational, and family support, the more likely they will support other members of the organization, improve job engagement and discipline, increase their sense of trust in the health care institution, and promote improved well-being at work, which contributes to decrease absenteeism, dismissal, and burnout.

In this sense, there is a strategic need allied with the PHC work process for health workers to negotiate in favor of well-being at work, promote improved commitment and job satisfaction, and establish positive bonds with the organization. Affective commitment motivates workers and promotes greater job involvement, which favors healthy environments. Thus, meetings should be held with managers/supervisors and workers, community members, and the occupational safety committee to negotiate improvements in the working conditions of PHC environments. Studies conducted in the United Kingdom [46,71] and Australia [72] on the well-being of health professionals related to the COVID-19 pandemic showed that management was involved in the negotiation process to favor positive bonds of job satisfaction and commitment to the health institution. Even though it depicted an extreme situation during the COVID-19 pandemic, management remains responsible for negotiating improved working conditions in favor of workers, even in adverse and conflicting situations that lead to negative and unpleasant job involvement. A study in Australia addressing the well-being of primary care nurses when facing the COVID-19 pandemic showed that being exposed to various personal and professional stressors affected the workers’ psychological well-being. Such a fact contributed to workers considering commitment and job satisfaction to be negatively tied to the health institution because the relationship established with PHC management was seen as a constructive relational obstacle at work. This is because interactions were not always effective when managing occupational risks and the quality of PHC services to meet the work and community needs. The healthcare systems were overwhelmed, dealing with changing healthcare demands such as increased patient care, suboptimal communication of infection control guidelines, high patient mortality, inadequate PPE supply, and fear of transmitting COVID-19 to family members. These factors harmed the health professionals’ well-being, commitment, job satisfaction, and involvement with PHC services.

However, the results of a study conducted in Italy about risk and protective factors of health professionals’ well-being are similar to those we report here regarding the interpersonal relationships of professionals supported by supervisors in negotiations intended to solve demands and establish safe working conditions. It appears as a positive factor in improving commitment and job satisfaction; i.e., it produces positive bonds with the health institution. The study also adds that identifying these protective factors of well-being at work, characteristics of environmental conditions, and positive individual behaviors strengthen job involvement as pleasant hours are experienced at work [73].

The reasons for negotiations promoted by the multidisciplinary team, such as redistributing working time, increased commitment and positive affective satisfaction at work, and promoting workers’ involvement with work in PHC, are protective factors contributing to health professionals’ subjective well-being.

Studies emphasize the importance of negotiating work shifts in health services as a strategic protective factor for coping with stress during the COVID-19 pandemic. The results show that stress was significantly decreased, revealing that negotiations were a fundamental protective factor for health professionals’ job engagement [74,75]. Furthermore, other studies show that the support provided by the professional network and especially from the direct manager in reshuffling work shifts was a vital factor favoring commitment and job satisfaction, in addition to job involvement, as these contribute to an improvement in workers’ moods and the planning of work tasks [74,76,77].

Well-being at work refers to health professionals’ positive mental state toward their performance in the work environment. The results showed that working conditions influence health professionals’ commitment, satisfaction, and job involvement, reaffirming the use of well-being at work when performing an organizational diagnosis and contributing to studies and the management of PHC services. These workers must perform quickly and efficiently under increasing pressure, which affects their health. Thus, the importance of well-being at work resides in the fact that the more committed, satisfied, and involved with work, the greater workers’ well-being, especially considering these workers’ potential and socio-environmental integration consisting of PHC work and broader social relations, such as public policies related to the organization of the work itself and the workforce with its needs, as well as the different social groups they assist.

### 5.1. Implications for PHC Policy and Management

Well-being at work can be a protective factor, promote health, and prevent distress, illness, inequity, vulnerability, and degradation of the work environment. In this sense, governance from the well-being at work perspective must direct human resources management toward strategies that promote the potential of workers and managers to negotiate sustainable working conditions to produce high levels of well-being at work.

Some elements that integrate work processes include the organization of specific tasks shared by teams to promote harmony and integration among workers; the control and assessment of workers’ exposure to occupational risks; individual behaviors (co-workers, managers, community) intended to change or minimize barriers and facilitate well-being in work processes; and inclusive governance to integrate representatives of stakeholders on decision-making concerning guidelines and standards regarding work organization at PHC.

Therefore, some operational strategies stand out in the organization of labor and management of the workforce: flexible working hours that enable workers to perform the various activities developed in PHC services remotely; health-promotion programs to train and qualify leaders and workers to manage the staff’s stressful conditions; providing psychological support and monitoring workers’ health conditions by implementing health tests and screenings regularly; and facilitating access to healthcare, including integrative and complementary practices.

Additionally, strategies should be devised to encourage PHC workers to individually and/or collectively participate in health policies aiming for well-being at work, including identifying, challenging, and assessing working conditions that put workers’ physical, mental, affective, or moral integrity at risk. Hence, workers should become security guards working with labor unions or other representative bodies to report irregularities to health surveillance and labor inspections services and collectively participate in associations, federations, unions, and other organizations to develop a priority agenda, dialoguing with public agencies responsible for health policies and devising actions intended to promote the well-being of workers.

We understand that well-being at work is a complex phenomenon with multiple aspects constantly challenging researchers and those who operationalize changes in the work routine. In this sense, researchers from the universities included in this and future studies are attentive and collaborate with the workers and managers at the local health network to dialogue about the planning and operationalization of interventions intended to promote well-being at work within PHC.

### 5.2. Limitations and Lines of Research

Regarding this study’s limitations, the cross-sectional design does not allow for causal relationships; hence, future studies should include longitudinal designs. Additionally, although an appropriate sample size was adopted, intentional sampling was used, and the study was performed in only two cities. Therefore, caution is needed to interpret and generalize the results [36]. Another limitation is that few studies address this population, which impedes comparisons between well-being at work indexes and the development of a historical series. Finally, although this study was conducted before the COVID-19 pandemic, low levels of well-being at work were identified. In this sense, this study can guide future studies and support an in-depth analysis of the movement of workers and managers toward negotiations to improve working conditions and well-being at work within a PHC facility.

## 6. Conclusions

These analysis results indicate that the professionals of a multi-professional PHC team present low levels of well-being at work, especially regarding job involvement. Additionally, nursing workers (technicians and nurses) more frequently expressed job commitment and job satisfaction. Our findings show that difficulties in solving problems, performing work routines, and co-workers’ attitudes directly influence the well-being of PHC team members. Risk perception (physical and chemical) also influences well-being. Moreover, negotiations in which PHC managers engage to improve working conditions are a significant predictor of job commitment, job satisfaction, and job involvement, especially when the reason to negotiate is redistributing working schedules.

These results show evidence that well-being scores increase when individual and collective working conditions are improved with the participation of managers. This study contributes to understanding well-being as a determinant factor of healthy work environments in PHC services. Additionally, it supports participatory negotiation strategies from the well-being perspective to improve working conditions with managers, policymakers, and workers.

## Figures and Tables

**Figure 1 ijerph-19-15943-f001:**
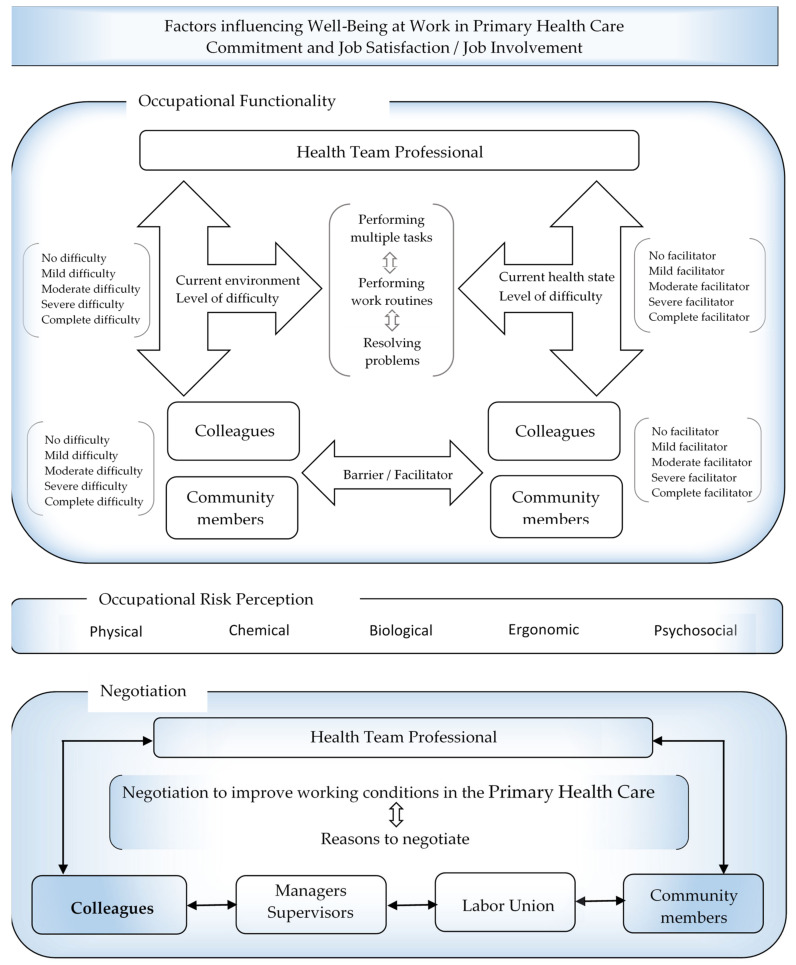
An outline of the main concepts integrated into Well-being at Work (occupational functionality, risk perception, and negotiation to improve working conditions) and respective qualifiers to investigate the magnitude of the problem among the professionals in the multidisciplinary PHC health teams.

**Figure 2 ijerph-19-15943-f002:**
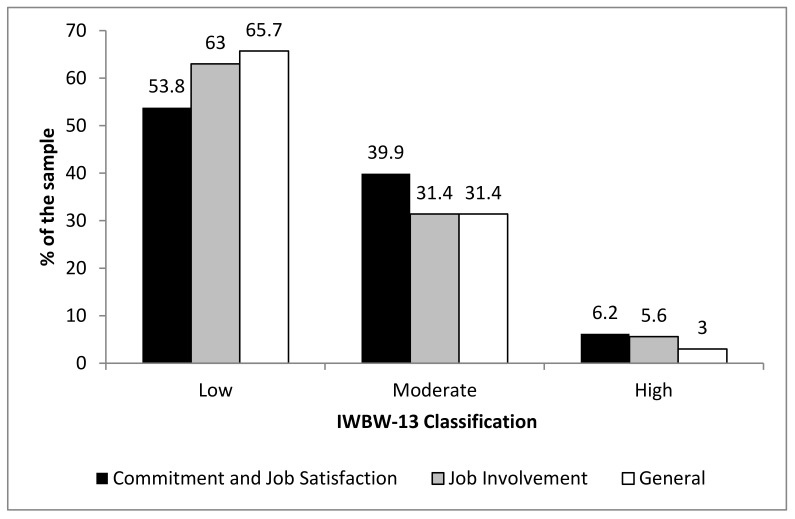
Assessment of the classification of the Inventory Well-Being at Work—(IWBW-13) scores.

**Table 1 ijerph-19-15943-t001:** Association between socio-demographic and job-related variables in the dimensions of well-being at work (Commitment and Satisfaction, Job Involvement).

Variables	n (%)	Commitment and Satisfaction	*p*	Job Involvement	*p*-Value
		Mean ± SD		Mean ± SD	
		2.83 ± 0.72		2.65 ± 0.72	
Age (years) *	41.4 ± 9.9	r = 0.088	0.106 ^a^	r = 0.068	0.215 ^a^
Sex			0.067 ^c^		0.016 ^c^
Male	45 (13.4)	3.01 ± 0.69		2.89 ± 0.62	
Female	292 (86.6)	2.80 ± 0.73		2.61 ± 0.73	
Race			0.653 ^b^		0.391 ^b^
Caucasian	257 (76.9)	2.81 ± 0.73		2.61 ± 0.73	
Afro-descendant	37 (11.1)	2.88 ± 0.70		2.81 ± 0.63	
Asian	1 (0.3)	2.11 ± 0.00		3.00 ± 0.00	
Mixed race	39 (11.7)	2.89 ± 0.69		2.70 ± 0.73	
Marital Status			0.038 ^b^		0.074 ^b^
Single	99 (29.3)	2.76 ± 0.73 ^ef^		2.66 ± 0.77	
Married/Consensual union	198 (58.6)	2.88 ± 0.71 ^ef^		2.68 ± 0.70	
Separate/Divorced	36 (10.7)	2.67 ± 0.70 ^e^		2.39 ± 0.67	
Widowed	5 (1.5)	3.53 ± 0.84 ^f^		3.10 ± 1.10	
Educational Level			0.827 ^b^		0.322 ^b^
Incomplete Middle/High School	7 (2.1)	2.89 ± 0.68		2.64 ± 0.80	
Complete High School	75 (22.2)	2.83 ± 0.69		2.62 ± 0.69	
Vocational School	47 (13.9)	2.90 ± 0.67		2.85 ± 0.80	
Some undergraduate studies	39 (11.5)	2.88 ± 0.83		2.72 ± 0.76	
Bachelor’s degree	92 (27.2)	2.72 ± 0.73		2.51 ± 0.68	
Technologist	6 (1.8)	2.89 ± 0.92		2.71 ± 0.43	
Specialization	56 (16.6)	2.91 ± 0.73		2.65 ± 0.72	
Master’s degree/Ph.D.	16 (4.7)	2.80 ± 0.78		2.77 ± 0.83	
Number of children **	1 (0–2)	r_s_ = 0.014	0.792 ^d^	r_s_ = −0.048	0.384 ^d^
Monthly Income (MW)			0.875 ^b^		0.789 ^b^
Up to 2 times the MW	130 (39.2)	2.80 ± 0.69		2.68 ± 0.75	
2 to 4 times the MW	115 (34.6)	2.79 ± 0.77		2.59 ± 0.72	
4 to 6 times the MW	21 (6.3)	2.97 ± 0.66		2.63 ± 0.69	
6 to 8 times the MW	18 (5.4)	2.90 ± 0.63		2.54 ± 0.57	
8 to 10 times the MW	21 (6.3)	2.75 ± 0.72		2.64 ± 0.73	
>10 times the MW	27 (8.1)	2.87 ± 0.78		2.80 ± 0.74	
City			0.124 ^c^		0.077 ^c^
Rio Grande	282 (83.4)	2.86 ± 0.72		2.62 ± 0.70	
São José do Norte	56 (16.6)	2.69 ± 0.75		2.80 ± 0.81	
Second job			0.512 ^c^		0.010 ^c^
Yes	62 (18.4)	2.88 ± 0.66		2.86 ± 0.68	
No	275 (81.6)	2.81 ± 0.73		2.60 ± 0.73	
UBS type			0.902 ^b^		0.605 ^b^
Traditional BHU	25 (7.4)	2.84 ± 0.85		2.87 ± 0.92	
Fluvial BHU	204 (60.4)	2.82 ± 0.68		2.62 ± 0.71	
Fluvial–Family Health Support	49 (14.5)	2.92 ± 0.82		2.61 ± 0.72	
24 h BHU	34 (10.1)	2.72 ± 0.87		2.75 ± 0.72	
Mixed Fluvial BHU	24 (7.1)	2.88 ± 0.58		2.59 ± 0.67	
Mobile BHU	2 (0.6)	2.89 ± 0.00		2.63 ± 0.53	
Profession			0.085 ^b^		0.005 ^b^
Nurse	50 (14.8)	2.91 ± 0.71		2.68 ± 0.67 ^ef^	
Physician	43 (12.7)	2.64 ± 0.80		2.59 ± 0.77 ^ef^	
Nursing technician/assistant	72 (21.3)	2.97 ± 0.70		2.93 ± 0.69 ^f^	
Community health agent	139 (41.1)	2.75 ± 0.70		2.50 ± 0.69 ^e^	
Dentist	13 (3.8)	3.14 ± 0.74		2.81 ± 0.74 ^ef^	
Oral health technician/assistant	15 (4.4)	2.93 ± 0.69		2.63 ± 0.93 ^ef^	
Other	6 (1.8)	2.63 ± 0.81		2.50 ± 0.63 ^ef^	
Years in the profession **	11 (3–16)	r_s_ = 0.054	0.327 ^d^	r_s_ = 0.020	0.721 ^d^
Years working in PHC **	8 (1–16)	r_s_ = −0.076	0.166 ^d^	r_s_ = −0.116	0.033 ^d^
Weekly hours	39.6 ± 5.3	r = −0.073	0.181 ^a^	r = −0.024	0.655 ^a^
Work shift at the PHC facility			0.296 ^b^		0.176 ^b^
Daytime	300 (89.3)	2.84 ± 0.71		2.64 ± 0.72	
Night-time	8 (2.4)	3.01 ± 1.00		3.09 ± 0.74	
Night-time/Daytime	23 (6.8)	2.67 ± 0.80		2.74 ± 0.75	
Other	5 (1.5)	2.38 ± 0.36		2.25 ± 0.47	
Physical Occupational Risk			0.012 ^c^		0.020 ^c^
No	31 (9.3)	3.14 ± 0.78		2.94 ± 0.71	
Yes	304 (90.7)	2.79 ± 0.71		2.62 ± 0.72	
Chemical Occupational Risk			0.006 ^c^		0.003 ^c^
No	65 (19.5)	3.05 ± 0.73		2.89 ± 0.71	
Yes	269 (80.5)	2.77 ± 0.72		2.59 ± 0.72	
Biological Occupational Risk			0.353 ^c^		0.187 ^c^
No	20 (5.9)	2.97 ± 0.77		2.85 ± 0.69	
Yes	317 (94.1)	2.82 ± 0.72		2.63 ± 0.72	
Ergonomic Occupational Risk			0.040 ^c^		0.064 ^c^
No	41 (12.2)	3.04 ± 0.76		2.84 ± 0.71	
Yes	296 (87.8)	2.79 ± 0.71		2.62 ± 0.72	
Psychosocial Occupational Risk			0.090 ^c^		0.001 ^c^
No	13 (3.8)	3.16 ± 0.85		3.29 ± 0.97	
Yes	325 (96.2)	2.82 ± 0.72		2.62 ± 0.70	

^a^ Pearson correlation coefficient; ^b^ Analysis of Variance (ANOVA); ^c^ Student’s *t*-test; ^d^ Spearman correlation coefficient; * described according to mean ± SD; ** described according to the median (percentiles 25–75); ^e,f^ No differences according to Tukey’s test at 5% level.

**Table 2 ijerph-19-15943-t002:** Association of the variables of worker functionality in the ICF with the dimensions of Well-being at Work (Commitment and Satisfaction, Job Involvement).

Variables	n (%)	Commitment and Satisfaction	*p*	Job Involvement	*p*-Value
		Mean ± SD		Mean ± SD	
Health State: Level of difficulty in solving problems **		r_s_ = −0.276	<0.001 ^d^	r_s_ = −0.162	0.003 ^d^
No difficulty	146 (43.2)				
Mild difficulty	100 (29.6)				
Moderate difficulty	83 (24.6)				
Severe difficulty	8 (2.4)				
Complete difficulty	1 (0.3)				
Current environment: Level of difficulty in solving problems **		r_s_ = −0.347	<0.001 ^d^	r_s_ = −0.249	<0.001 ^d^
No difficulty	93 (27.6)				
Mild difficulty	106 (31.5)				
Moderate difficulty	119 (35.3)				
Severe difficulty	12 (3.6)				
Complete difficulty	7 (2.1)				
Health state: Level of difficulty in performing multiple tasks **		r_s_ = −0.211	<0.001 ^d^	r_s_ = −0.180	0.001 ^d^
No difficulty	135 (39.9)				
Mild difficulty	112 (33.1)				
Moderate difficulty	72 (21.3)				
Severe difficulty	18 (5.3)				
Complete difficulty	1 (0.3)				
Current environment: Level of difficulty in performing multiple tasks **		r_s_ = −0.323	<0.001 ^d^	r_s_ = −0.255	<0.001 ^d^
No difficulty	110 (32.6)				
Mild difficulty	100 (29.7)				
Moderate difficulty	102 (30.3)				
Severe difficulty	21 (6.2)				
Complete difficulty	4 (1.2)				
Health state: Level of difficulty in performing work routines **		r_s_ = −0.274	<0.001 ^d^	r_s_ = −0.266	<0.001 ^d^
No difficulty	165 (49.0)				
Mild difficulty	95 (28.2)				
Moderate difficulty	59 (17.5)				
Severe difficulty	17 (5.0)				
Complete difficulty	1 (0.3)				
Current environment: Level of difficulty in performing work routines **		r_s_ = −0.332	<0.001 ^d^	r_s_ = −0.282	<0.001 ^d^
No difficulty	118 (34.9)				
Mild difficulty	112 (33.1)				
Moderate difficulty	87 (25.7)				
Severe difficulty	20 (5.9)				
Complete difficulty	1 (0.3)				
Scores assigned to the community’s attitudes: Barriers **		r_s_ = −0.198	<0.001 ^d^	r_s_ = −0.110	0.044 ^d^
No barriers	105 (31.2)				
Mild barriers	102 (30.3)				
Moderate barriers	106 (31.5)				
Severe barriers	20 (5.9)				
Complete barriers	4 (1.2)				
Scores assigned to the community’s attitudes: Facilitators **		r_s_ = 0.065	0.239 ^d^	r_s_ = 0.076	0.172 ^d^
No facilitator	67 (20.4)				
Mild facilitator	90 (27.4)				
Moderate facilitator	99 (30.2)				
Substantial facilitator	56 (17.1)				
Complete facilitator	16 (4.9)				
Scores assigned to the colleagues’ attitudes: Barriers **		r_s_ = −0.275	<0.001 ^d^	r_s_ = −0.101	0.063 ^d^
No barriers	140 (41.4)				
Mild barriers	87 (25.7)				
Moderate barriers	83 (24.6)				
Severe barriers	23 (6.8)				
Complete barriers	5 (1.5)				
Scores assigned to the colleagues’ attitudes: Facilitators **		r_s_ = −0.027	0.625 ^d^	r_s_ = 0.063	0.252 ^d^
No facilitator	54 (16.1)				
Mild facilitator	78 (23.3)				
Moderate facilitator	77 (23.0)				
Substantial facilitator	87 (26.0)				
Complete facilitator	39 (11.6)				

^d^ Spearman’s correlation coefficient; ** Described according to the medians (percentiles 25–75). The concepts provided by the International Classification of Functioning, Disability and Health (ICF) concerning health state, current environment, and barriers and facilitators presented by community members and colleagues were used.

**Table 3 ijerph-19-15943-t003:** Associations between Worker Negotiation for improving working conditions in PHC services (with whom/source and for what/reasons) variables with the dimensions of Well-being at Work (Commitment and Satisfaction, Job Involvement).

Variables	n (%)	Commitment and Satisfaction	*p*	Job Involvement	*p*-Value
		Mean ± SD		Mean ± SD	
Negotiates to improve the PHC working conditions w/workers/colleagues			0.003 ^c^		0.038 ^c^
No	143 (42.6)	2.97 ± 0.72		2.75 ± 0.78	
Yes	193 (57.4)	2.73 ± 0.71		2.58 ± 0.67	
Negotiates to improve the PHC working conditions w/labor union			0.360 ^c^		0.027 ^c^
No	303 (90.2)	2.85 ± 0.72		2.68 ± 0.73	
Yes	33 (9.8)	2.72 ± 0.73		2.39 ± 0.60	
Negotiates to improve the PHC working conditions w/managers/supervisors			<0.001 ^c^		<0.001 ^c^
No	182 (54.2)	2.61 ± 0.68		2.48 ± 0.65	
Yes	154 (45.8)	3.10 ± 0.68		2.85 ± 0.76	
Negotiates to improve the PHC working conditions w/community members			0.001 ^c^		0.656 ^c^
No	303 (90.2)	2.79 ± 0.72		2.65 ± 0.72	
Yes	33 (9.8)	3.24 ± 0.61		2.70 ± 0.74	
Negotiates to improve the PHC working conditions w/occupational safety workers			0.005 ^c^		0.027 ^c^
No	326 (97.0)	2.81 ± 0.72		2.64 ± 0.72	
Yes	10 (3.0)	3.47 ± 0.65		3.15 ± 0.61	
Reasons to negotiate: Change the conditions of exposure to physical risks			0.519 ^c^		0.617 ^c^
No	185 (56.7)	2.87 ± 0.72		2.64 ± 0.68	
Yes	141 (43.3)	2.82 ± 0.72		2.68 ± 0.78	
Reasons to negotiate: Change the conditions of exposure to chemical risks			0.292 ^c^		0.995 ^c^
No	270 (82.8)	2.86 ± 0.69		2.66 ± 0.72	
Yes	56 (17.2)	2.75 ± 0.80		2.66 ± 0.74	
Reasons to negotiate: Change the conditions of exposure to biological risks			0.397 ^c^		0.990 ^c^
No	234 (71.8)	2.87 ± 0.69		2.66 ± 0.71	
Yes	92 (28.2)	2.79 ± 0.77		2.65 ± 0.76	
Reasons to negotiate: Change the conditions of exposure to ergonomic risks			0.041 ^c^		0.591 ^c^
No	213 (65.3)	2.90 ± 0.71		2.67 ± 0.70	
Yes	113 (34.7)	2.73 ± 0.71		2.63 ± 0.76	
Reasons to negotiate: Change the conditions of exposure to occupational accidents			0.010 ^c^		0.045 ^c^
No	248 (76.1)	2.90 ± 0.69		2.70 ± 0.71	
Yes	78 (23.9)	2.66 ± 0.77		2.51 ± 0.74	
Reasons to negotiate: Change the conditions of exposure to psychosocial risks			0.001 ^c^		0.040 ^c^
No	188 (57.7)	2.96 ± 0.71		2.73 ± 0.74	
Yes	138 (42.3)	2.69 ± 0.69		2.56 ± 0.68	
Reasons to negotiate: Arrange the acquisition of PPEs			0.049 ^c^		0.237 ^c^
No	192 (58.9)	2.91 ± 0.69		2.69 ± 0.71	
Yes	134 (41.1)	2.75 ± 0.74		2.60 ± 0.74	
Reasons to negotiate: Encourage employee compliance w/PPE			0.815 ^c^		0.517 ^c^
No	210 (64.4)	2.85 ± 0.71		2.67 ± 0.73	
Yes	116 (35.6)	2.83 ± 0.73		2.62 ± 0.71	
Reasons to negotiate: Increase salaries			0.001 ^c^		0.828 ^c^
No	131 (40.2)	3.00 ± 0.70		2.65 ± 0.72	
Yes	195 (59.8)	2.74 ± 0.71		2.66 ± 0.73	
Reasons to negotiate: Discuss/change work goals			0.086 ^c^		0.076 ^c^
No	137 (42.0)	2.93 ± 0.71		2.74 ± 0.69	
Yes	189 (58.0)	2.79 ± 0.71		2.59 ± 0.74	
Reasons to negotiate: Resolve conflicts of interest			<0.001 ^c^		0.051 ^c^
No	159 (48.8)	3.00 ± 0.68		2.74 ± 0.67	
Yes	167 (51.2)	2.70 ± 0.72		2.58 ± 0.76	
Reasons to negotiate: Promote educational qualification			0.173 ^c^		0.633 ^c^
No	163 (50.0)	2.90 ± 0.69		2.67 ± 0.72	
Yes	163 (50.0)	2.79 ± 0.73		2.64 ± 0.73	
Reasons to negotiate: Redistribute working shifts			0.042 ^c^		0.034 ^c^
No	289 (88.7)	2.88 ± 0.69		2.69 ± 0.72	
Yes	37 (11.3)	2.57 ± 0.87		2.42 ± 0.69	
Reasons to negotiate: Obtain the right to health at work			0.135 ^c^		0.637 ^c^
No	209 (64.1)	2.89 ± 0.69		2.67 ± 0.71	
Yes	117 (35.9)	2.77 ± 0.76		2.63 ± 0.76	
Reasons to negotiate: Working hours			0.022 ^c^		0.608 ^c^
No	227 (69.6)	2.91 ± 0.68		2.64 ± 0.71	
Yes	99 (30.4)	2.71 ± 0.77		2.69 ± 0.75	
Reasons to negotiate: Improve interaction/communication among workers			0.213 ^c^		0.707 ^c^
No	218 (66.9)	2.88 ± 0.69		2.67 ± 0.69	
Yes	108 (33.1)	2.77 ± 0.76		2.63 ± 0.77	
Reasons to negotiate: Improve interaction/communication between workers and employers			0.430 ^c^		0.801 ^c^
No	214 (65.6)	2.87 ± 0.69		2.65 ± 0.69	
Yes	112 (34.4)	2.80 ± 0.77		2.67 ± 0.78	
Reasons to negotiate: Discuss the resources provided to labor unions			0.164 ^c^		0.352 ^c^
No	300 (92.0)	2.86 ± 0.71		2.67 ± 0.72	
Yes	26 (8.0)	2.66 ± 0.80		2.53 ± 0.76	

^c^ Student’s *t*-test.

**Table 4 ijerph-19-15943-t004:** Multivariate Linear Regression Analysis to assess factors independently associated with the dimension Commitment and Satisfaction of well-being at work.

Factors	b (95% CI)	Beta	*p*	R^2^
Commitment and Satisfaction				36.4%
Place: Rio Grande	0.26 (0.08–0.44)	0.138	0.005	
Profession				
Nurse	0.31 (0.11–0.51)	0.152	0.003	
Physician	−0.23 (−0.44–−0.01)	−0.105	0.040	
Nursing technician/Assistant	0.27 (0.10–0.45)	0.156	0.002	
Community health agent	0.00			
Dentist	0.42 (0.08–0.76)	0.117	0.015	
Oral health technician/assistant	0.12 (−0.24–0.47)	0.032	0.519	
Other	0.12 (−0.40–0.64)	0.022	0.656	
Years working in PHC services	−0.01 (−0.02–−0.00)	−0.107	0.027	
Physical occupational risk	−0.32 (−0.56–−0.08)	−0.132	0.010	
Chemical occupational risk	−0.21 (−0.39–−0.03)	−0.118	0.021	
Current environment: Difficulty level in solving problems	−0.15 (−0.23–−0.07)	−0.197	<0.001	
Scores assigned to colleagues’ attitudes: Barrier	−0.09 (−0.16–−0.02)	−0.124	0.017	
Negotiating for improvements w/managers/supervisors	0.44 (0.31–0.58)	0.308	<0.001	
Negotiating for improvements w/community members	0.40 (0.17–0.63)	0.168	0.001	
Reasons for negotiation: Redistribution of working shifts	−0.23 (−0.44–−0.02)	−0.104	0.032	

b = regression coefficient; 95% CI = 95% confidence interval; Beta = standardized regression coefficient; R^2^ = determination coefficient.

**Table 5 ijerph-19-15943-t005:** Multivariate Linear Regression Analysis to assess factors independently associated with the Job Involvement dimension of Work Well-being.

Factors	b (95% CI)	Beta	*p*	R^2^
Job Involvement				24.4%
Profession				
Nurse	0.21 (−0.01 to 0.42)	0.099	0.057	
Physician	0.04 (−0.20 to 0.28)	0.018	0.744	
Nursing technician/Assistant	0.46 (0.28 to 0.64)	0.260	<0.001	
Community health agent	0.00			
Dentist	0.33 (−0.04 to 0.69)	0.089	0.080	
Oral health technician/Assistant	0.06 (−0.31 to 0.43)	0.017	0.745	
Other	−0.08 (−0.62 to 0.47)	−0.014	0.785	
Work shift				
Daytime	0.45 (−0.13 to 1.02)	0.188	0.127	
Night-time	0.77 (0.03 to 1.52)	0.156	0.042	
Night-time/Daytime	0.71 (0.08 to 1.34)	0.243	0.028	
Other	0.00			
Chemical occupational hazard	−0.32 (−0.51 to −0.14)	−0.176	0.001	
Current environment: Level of difficulty in performing work routines	−0.17 (−0.25 to −0.09)	−0.218	<0.001	
Negotiation to improve working conditions: managers/supervisors	0.39 (0.25 to 0.54)	0.272	<0.001	
Reasons for negotiation: Redistribution of working shifts	−0.25 (−0.48 to −0.03)	−0.111	0.030	

b = regression coefficient; 95% CI = 95% confidence interval; Beta = standardized regression coefficient; R^2^ = coefficient of determination.

## Data Availability

Data regarding this study can be provided upon request to the corresponding author. Data are not publicly available due to ethical issues.

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
