# Peer review of "Occupational Well-Being of Multidisciplinary PHC Teams: Barriers/Facilitators and Negotiations to Improve Working Conditions"

_ijerph, 2022, doi:10.3390/ijerph192315943_

Round 1

Reviewer 1 Report

Dear authors,

Congratulations on your massive project in the south of Brazil, which has already resulted in several publications.

This time, the work is about the negotiations regarding the working conditions.

It made it difficult for me to evaluate the paper because it referred to literature five and, in addition, several articles by the same author. The scientific results presented in this article are thus not directly supported, as we should be familiar with previous works. I consider it essential that the publication contains the problem statement, the literature review, and the background of the applied research plan and approach concept directly. It would be better if the reference to the fifth publication were significantly smaller.

In addition to the IWBW-13 tool containing 13 questions, the article uses a questionnaire about demographics, workplace conditions and negotiations.

I am missing from the article the evaluation of the IWBW-13 tool and the confirmation that the two scales with the 9+3 items are reliable in this case.

I suggest shortening the paper and putting Figure 1 in focus.

Author Response

Response to Reviewer 1 Comments

 Dear Reviewer,

We greatly appreciate your comments. We carried out the recommendations and made improvements to the manuscript. We are writing to provide peer-to-peer responses to your comments.

Comments and Suggestions for Authors:

Congratulations on your massive project in the south of Brazil, which has already resulted in several publications.

This time, the work is about the negotiations regarding the working conditions.

Point 1: It made it difficult for me to evaluate the paper because it referred to literature five and, in addition, several articles by the same author.

Response 1: We appreciate your comment.  We have corroborated your comment and improved and reorganized the introduction to address the recommendations. We emphasize that the concept of Well-being at Work is supported by processes theorized by authors presented in the introduction, framework, and method. From which the different metrics were elaborated on and built over time in different surveys. In our study, we used one of them (metric) that was previously validated and applied in different work environments (private and public services) in Brazil. In this academic sense, our study can contribute to its visualization and evaluation in the scientific community that studies the theme of Well-being at work. We aligned the statements made in the introduction with references that allowed maintaining the coherence and adequacy of the content of this manuscript. Other aspects were strengthened in the method, as can be seen following this response report.

Point 2: The scientific results presented in this article are thus not directly supported, as we should be familiar with previous works.

Response 2: We appreciate the comment and highlight the studies used to identify similarities and differences in the research problem. Well-being at work and its influencers, for which other concepts referred to in the global framework, were included: occupational functionality, risk perception, and negotiation for improvements in working conditions, in addition to the sociodemographic and occupational characteristics of the group of health workers who participated in this study. We have included other studies in English to expand the dialogue between those interested in the subject. We welcome the following Brazilian investigations versed in English: Abreu, H.D.; Blanco, A.J.M.R. The well-being at work and resilience: a study correlation in nursing technicians in hospital. Rev Psicol Divers Saúde. 2017, 6, 170-180. Available online: https://doi.org/10.17267/2317-3394rpds.v6i3.1434 (accessed on 10 September 2022).

Libardi, M.B.O.; Arrais, A.R.; Antloga, C.S.X.; Faiad, C.; Rodrigues, C.M.L.; Barros, A.F. Gender, psychosocial stressors, well-being and coping in prehospital care workers. Rev Bras Enferm. 2021.74(Suppl 3), e20200579. doi: https://doi.org/10.1590/0034-7167-2020-0579

Point 3: I consider it essential that the publication contains the problem statement, the literature review, and the background of the applied research plan and approach concept directly. It would be better if the reference to the fifth publication were significantly smaller.

Response 3: We understand your placements. We made improvements and adjustments to the manuscript. The concept of well-being at work is complex. Different approaches are used to revitalize the psychological meaning of a positive cognitive and emotional state of the person at work, which can be a driver of attitudes and, thus, promote a state of collective sensation in the environment. of work. The guiding concepts of the present manuscript were described and presented, and in the discussion, we approached studies and reflections that were current with the present study. The authors improved the explanation in the framework and throughout the manuscript (introduction, method, and discussion).

They used different references to demonstrate the relevance of the study of well-being at work. Yes, we carefully review and replace and include other references that contribute to the elucidation and discussion of the research problem.

We added a sub-item to the Discussion section (5.1. Implications for PHC policy and management) on page 25, lines 964 to 1003.

We hope that we have reached the recommendations.

Point 4: In addition to the IWBW-13 tool containing 13 questions, the article uses a questionnaire about demographics, workplace conditions and negotiations.

Response 4: We appreciate the comment. We want to inform you that we have translated into English the Inventory of Well-being at Work and the selected questions from the more extensive questionnaire used in this manuscript (sociodemographic and occupational characteristics). These are shown under Supplementary Materials (Figure S1 and Figure S2, respectively). Please note that the IWBW-13 is not a validated scale in English and is for informational purposes only. As well as, the questions used in the project's more extensive questionnaire do not constitute a validated questionnaire in English and are for informational purposes only.

Point 5: I am missing from the article the evaluation of the IWBW-13 tool and the confirmation that the two scales with the 9+3 items are reliable in this case.

Response 5: We appreciate the recommendation. The authors proceeded to include the evaluation of the tool in the present manuscript. We added the information in Materials and Methods as follows:

“Given a consensus in the literature addressing psychometric properties that Cronbach’s alpha is not recommended if there is uncertainty regarding an instrument’s one dimension or suspected multi-dimension, experts suggest that McDonald’s omega be used to assess an instrument’s reliability. This coefficient is recommended due to its good performance in breaking assumptions for Cronbach’s alpha [Green, S.B. and Yang, Y., 2015]. In this study, the McDonald’s omega obtained by the Commitment and Satisfaction Factor was 0.82, the Job Involvement Factor obtained 0.65, and the total scale 0.86. These reliability measures are usually above 0.7; thus, only the Job Involvement Factor obtained a lower score. However, according to Altman [1991], a reliability coefficient above 0.6 would already be satisfactory, considering the Job Involvement factor presents a lower number of items (4). Thus, the scale presented good internal consistency.” (page 9, lines 406 to 418)

Green, S. B., & Yang, Y. (2015). Evaluation of Dimensionality in the Assessment of Internal Consistency Reliability: Coefficient Alpha and Omega Coefficients. Educational Measurement: Issues and Practice, 34(4), 14–20. doi:10.1111/emip.12100

Altman, D.G. Practical Statistics for Medical Research. Chapman & Hall. London, UK, 1991.

We have added Tables S1 and S2 in Supplementary Material; these present the complete Multivariate Linear Regression model for the Inventory Well-Being at Work scores [Table S1 Multivariate Linear Regression Analysis to assess factors independently associated with the Commitment and Satisfaction dimension of the Inventory Well-being at Work and Table S2 Multivariate Linear Regression Analysis to assess factors independently associated with the Job Involvement dimension of the Inventory Well-being at Work].

We have also included the SPSS Syntax program in PDF as an attachment to this response report.

Point 6: I suggest shortening the paper and putting Figure 1 in focus.

Response 6: We appreciate your recommendations. The manuscript was long in the face of our attempt to analyze occupational well-being (Commitment and Job Satisfaction, Job Involvement dimensions) by integrating the different concepts used in the study (occupational functionality, risk perception, negotiation for improvements in working conditions) as well as the sociodemographic and occupational characteristics of the workers of the PHC multi-professional team. We also inform you that we have reduced the presentation of results in agreement with you and another reviewer. We removed the entire part of the results entitled "Variables independently associated with the two factors Commitment and Job Satisfaction and Job Involvement." We evaluated and identified that the analysis developed in "4.2. Multivariate Linear Regression Analysis" was sufficient. In this way, the manuscript went from 34 pages to 31 pages. Thus, the text was shortened, with attention to maintaining the adequacy of the content.

We improved the quality of Figure 2.

We also inform that we separated the presentation of the results into sub-items, and we evaluated that it was consistent with what was proposed in the manuscript.

Also, at the end of the Results section, we have included a sub-item of Synthesis of the occupational well-being of the multidisciplinary PHC team.

We have improved and shortened the conclusion section of the study.

We inform you that it was a revision of the English language.

We inform you that there was a slight change in the writing in the title (Occupational well-being of multidisciplinary PHC teams: barriers/facilitators and negotiations to improve working conditions).

Thank you very much for your review work.

Reviewer 2 Report

I really appreciate the research presented and the quality of the overall paper. I would only suggest a further development in the discussion section concerning the implications for possible policy makers and HR management.

Author Response

Response to Reviewer 2 Comments

Dear Reviewer,

We greatly appreciate your comments. We carried out the recommendations and made improvements to the manuscript. We are writing to provide peer-to-peer responses to your comments.

Comments and Suggestions for Authors:

I really appreciate the research presented and the quality of the overall paper.

Point 1: I would only suggest a further development in the discussion section concerning the implications for possible policy makers and HR management.

Response 1: We appreciate your comment. We added a sub-item to the Discussion section (5.1. Implications for PHC policy and management) on page 25, lines 964 to 1003.

We also inform you that we have reduced the presentation of results in agreement with another reviewer. Thus, the text was shortened, with attention to maintaining the adequacy of the content.

We inform you that it was a revision of the English language.

We inform you that there was a slight change in the writing in the title (Occupational well-being of multidisciplinary PHC teams: barriers/facilitators and negotiations to improve working conditions).

Thank you very much for your review work.

Reviewer 3 Report

This is a nicely written manuscript on an important topic, wellbeing of workers and how to promote it. The manuscript is quite long, mainly because it try to cover work wellbeing as a whole and is based on global frameworks. The manuscript should gain of a shortening of the text. In addition, the analysis of results by profession could be leaved out, because of the rather small number of people (n=338) and therefore some small groups among the professions. Table 4 has no n? The first chapter of conclusions is long and cryptic? 

Author Response

Response to Reviewer 3 Comments

Dear Reviewer,

We greatly appreciate your comments. We carried out the recommendations and made improvements to the manuscript. We are writing to provide peer-to-peer responses to your comments.

Comments and Suggestions for Authors:

This is a nicely written manuscript on an important topic, wellbeing of workers and how to promote it. The manuscript is quite long, mainly because it try to cover work wellbeing as a whole and is based on global frameworks.

Point 1: The manuscript should gain of a shortening of the text.

Response 1: We appreciate your comment. The manuscript was long in the face of our attempt to analyze occupational well-being (Commitment and Job Satisfaction, Job Involvement dimensions) by integrating the different concepts used in the study (occupational functionality, risk perception, negotiation for improvements in working conditions) as well as the sociodemographic and occupational characteristics of the workers of the PHC multi-professional team. We tried to shorten the text a little from its evaluation and recommendation in Point 2; that is, we removed the entire part of the results entitled "Variables independently associated with the two factors Commitment and Job Satisfaction and Job Involvement." We evaluated and identified that the analysis developed in "4.2. Multivariate Linear Regression Analysis" was sufficient. In this way, the manuscript went from 34 pages to 31 pages.

We are very grateful for the recommendation.

Point 2: In addition, the analysis of results by profession could be leaved out, because of the rather small number of people (n=338) and therefore some small groups among the professions.

Response 2: We complied with your recommendation, which led us to carry out reappraise this part of the presentation of the results. We see that this is optional because of the previous results presented.

We also inform that we separated the presentation of the results into sub-items, and we evaluated that it was consistent with what was proposed in the manuscript.

We are very grateful for the recommendation.

Point 3: Table 4 has no n?

Response 3: We appreciate your comment. We justify that the "n" was not directly placed in the table because Table 4 presents the results of the "Multivariate Linear Regression Analysis to assess factors independently associated with the dimension Commitment and Satisfaction of Well-Being at Work." As the Multivariate Linear Regression analysis with the Backward model extraction method to control for confounding factors was used, we started with the total (n=338). Still, there were more than ten blocks for the final analysis. In this sense, for further information, we have added Tables S1 and S2 in Supplementary Material; these present the complete Multivariate Linear Regression model for the Inventory Well-Being at Work scores [Table S1 Multivariate Linear Regression Analysis to assess factors independently associated with the Commitment and Satisfaction dimension of the Inventory Well-being at Work and Table S2 Multivariate Linear Regression Analysis to assess factors independently associated with the Job Involvement dimension of the Inventory Well-being at Work]. We have also included the SPSS Syntax program in PDF as an attachment to this response report.

We are very grateful for the recommendation.

Point 4: The first chapter of conclusions is long and cryptic?

Response 4: We appreciate your recommendation. We have improved and shortened the conclusion section of the study.

We also inform you that we have added a sub-item in the discussion section [5.1. Implications for PHC policy and management] to better elucidate our projections articulated with local management and the workers of the PHC multi-professional team.

We inform you that it was a revision of the English language.

We inform you that there was a slight change in the writing in the title (Occupational well-being of multidisciplinary PHC teams: barriers/facilitators and negotiations to improve working conditions).

Thank you very much for your review work.
